# Effect of Breed and Season in Buck Semen Cryopreservation: The Portuguese Animal Germplasm Bank

**DOI:** 10.3390/vetsci11070326

**Published:** 2024-07-18

**Authors:** João Pedro Barbas, Maria Conceição Baptista, Nuno Carolino, João Simões, Gisele Margatho, Jorge Pimenta, Francisca Claudino, Filipa Costa Ferreira, Francisco Grilo, Rosa Maria Lino Neto Pereira

**Affiliations:** 1Department of Biotechnology and Genetic Resources, Instituto Nacional de Investigação Agrária e Veterinária, Quinta da Fonte Boa, 2005-424 Vale de Santarém, Portugal; pedro.barbas@iniav.pt (J.P.B.); batista.sao@gmail.com (M.C.B.); nuno.carolino@iniav.pt (N.C.); jorgepimenta7@gmail.com (J.P.); filipa.ferreira@iniav.pt (F.C.F.); rosa.linoneto@iniav.pt (R.M.L.N.P.); 2CIISA-AL4AnimalS-Faculty of Veterinary Medicina, Universidade de Lisboa, 1300-477 Lisboa, Portugal; 3Department of Veterinary Sciences, Veterinary and Animal Research Centre (CECAV), AL4AnimalS, School of Agricultural and Veterinary Sciences, University of Trás-os-Montes and Alto Douro (UTAD), 5000-801 Vila Real, Portugal; giselem@utad.pt; 4Instituto Nacional de Investigação Agrária e Veterinária, Innovation and Experimentation Hub, Quinta da Fonte Boa, 2005-424 Vale de Santarém, Portugal; francisca.claudino@iniav.pt (F.C.); francisco.grilo@iniav.pt (F.G.)

**Keywords:** spermatozoa, native goats, germplasm bank, reproductive biotechnologies

## Abstract

**Simple Summary:**

This study aimed to evaluate the effect of breed (four native Portuguese goat breeds), season (February–April, May–August, and September–January) and semen processing (fresh and cryopreserved semen) on semen traits. A mixed linear model with repeated measures and three-way interactions was used to test the volume of ejaculate, spermatozoa (SPZ) concentration, total SPZ per ejaculate, SPZ viability, individual motility, normal morphology, and head, midpiece, and tail defects of semen from 616 (Serrana), 207 (Bravia), 171 (Charnequeira), and 23 (Preta de Montezinho) buck ejaculates. Overall, a significant and consistent effect of breed, season, and their interactions were observed on ejaculate traits. In turn, these factors seemed to have a limited impact on the studied SPZ traits, contrarily to the semen processing. Despite some differences observed between breeds that require further research, this study indicates that these native goat breeds can be used as semen donors at our latitude during the whole year under natural conditions.

**Abstract:**

The aims of this study were to characterize the semen as well as the influence of breed, season, and semen processing on spermatozoa (SPZ) traits of four native Portuguese goat breeds used for the bank of Portuguese animal germplasm (BPAG). A total of 1017 ejaculates from Serrana (n = 30), Bravia (n = 15), Charnequeira (n = 11), and Preta de Montezinho (n = 3) bucks were collected between 2004 and 2020 at (EZN-INIAV; 39° N) during the whole year under natural conditions. All the fresh and cryopreserved (−196 °C) semen was evaluated and stored in the BPAG. Bravia bucks (the smallest breed) produced less (*p* < 0.05) volume of ejaculate than all the other breeds, which was higher during the full breeding season (September–January; *p* < 0.05), regarding all the other breeds. Contrarily, in general, SPZ concentration was lower during September–January, but total SPZ per ejaculate remained similar (*p* > 0.05) during May–August and September–January in Serrana bucks. The SPZ viability and SPZ midpiece defects were slightly influenced by breed and SPZ head defects by season (lowest % in February–April; *p* < 0.05). On the contrary, the freezing–thawing cycle strongly influenced (*p* < 0.01) all SPZ traits. The correlation coefficients of these traits between fresh and thawed SPZ were low (up to 0.33; *p* < 0.01), highlighting the importance of semen processing in semen cryopreservation. We conclude that breed and season had a relevant effect on ejaculate traits, but it was much less evident for the studied SPZ traits. These native goats can serve as semen donors throughout the year, under natural conditions.

## 1. Introduction

Portugal is a hub of animal biodiversity, housing 63 native Portuguese breeds of different domestic species. Native breeds of small ruminant species represent a valuable genetic heritage, which is essential for the biodiversity of the Portuguese rural landscape. They also represent an instrument of territorial cohesion and can mitigate human desertification, maintaining marginal agricultural areas with less fertile soils and preventing rural fires. Six native goat breeds—Bravia, Serrana, Charnequeira, Serpentina, Algarvia, and Preta de Montesinho—are recognized through the respective herdbook [1,2,3]. Serrana and Charnequeira breeds include at least two ecotypes, and all breeds are reared for a dual purpose (milk and meat), except Bravia goats (meat purpose). Native goats represent about 12.5% of all goats reared in Portugal [3]. These goat breeds are considered at risk of extinction, with the exception of Serrana goats (about 13,000 breeding animals). Currently, all native Portuguese goat breeds are all included in conservation or breeding programs with different objectives (meat, milk, or dual purpose).

The bank of Portuguese animal germplasm (BPAG) has been developed since 2004 at Estação Zootécnica National (EZN) to preserve reproductive cells from native breeds of different domestic animal species and ensure the conservation and maintenance of their genetic variability [4]. BPAG was officially established in 2011.

Native Portuguese goats are reared, mainly in mountain regions, in extensive or semi-extensive production systems. They are well adapted to harsh agroclimatic conditions, with extreme temperatures and complex orography. Most herds have fewer than 80 animals grazing on natural pastures and shrub vegetation. In more intensive production systems, they have access to dryland sown pastures. In periods of scarce or low-quality pastures, animals are supplemented with natural or sown fodders. Also, during the last third of pregnancy and during lactation, they are supplemented with high-nutritional-quality forages (e.g., vetch-oat hay, rye grass, sorghum, intercrops and legumes, oats and lupins, lupines, and triticale grasses) and fed with concentrates, depending on the prolificacy, season of parturition, and productivity level. Meat, milk, and cheese are still a major source of revenue for some rural populations. Most meat and cheese from native goats are certified products regarding the breed and production system.

At our latitude, native goat breeds show a moderate reproductive seasonality, which is evident in winter, and reported in female Serrana goats [5] and other native Portuguese ram breeds [6,7]. This reproductive pattern does not resemble that of goat breeds originating from higher latitudes [8], which show a marked seasonality with an anestrus season from late February to August. Usually, at our latitude and for native goats, spring is the main breeding season (May–June), with a second one in autumn (September–October), adjusting kid meat offer to market demand in Christmas and Easter with higher prices. The reproductive activity of small ruminants is influenced by genetic and environmental factors, namely photoperiod, but also nutrition, body condition, health status, temperature, and atmospheric humidity [9]. This influence is also reflected in Serrana bucks, where seasonal variations in semen quality can be detected throughout the year. A greater volume of ejaculate (VE), alive and motile spermatozoa (SPZ), and less SPZ abnormalities were observed during the breeding season, and a decrease in semen volume and increase in SPZ concentration were reported during spring and summer seasons [10]. In general, our native buck breeds exhibit low seasonality, having regular sexual activity (e.g., libido and mounting) throughout the year. In most flocks, natural mating lasts 45 days, entering the breeding season without a previous breeding soundness examination, namely andrological analysis, which can influence the fertility efficiency.

It is therefore imperative to improve reproductive flocks’ efficiency to increase farmers’ income and support ongoing conservation and breeding programs. The two major procedures that should be implemented are the characterization of ejaculates and spermatozoa (SPZ) traits in (1) fresh semen to be used for natural mating regarding the different seasons of semen collection of native goat breeds, and (2) after the freezing–thawing cycle for artificial insemination purposes. Semen cryopreservation is an assisted reproductive technology that allows for the preservation of animal genetic resources, while maintaining their viability and fertilization potential, if well-stored in nitrogen containers [4]. Despite the negative effect on the viability, functionality, and fertilization potential of SPZ, cryopreservation is a suitable technology to preserve semen, especially in males of endangered breeds and/or with high genetic value. Also, SPZ cryopreservation is essential for in situ or ex situ conservation after the donor’s death [11].

Regarding the context, the present study aimed to evaluate the influence of season on ejaculate and SPZ traits in fresh and thawed semen of four native Portuguese goat breeds stored in BPAG. For this purpose, an evaluation of the ejaculate and SPZ was carried out in fresh semen after each ejaculate collection and then after the freezing–thawing procedure, to ensure the minimum requirements for germplasm storage.

## 2. Materials and Methods

This study was approved by BPAG monitoring Committee and National Institute for Agricultural and Veterinary Research (INIAV), I.P.

### 2.1. Animals and Management

Four native Portuguese buck breeds—Bravia, Charnequeira, Preta de Montezinho, and Serrana—were used for semen collection between 2002 and 2020, throughout the year. Males (n = 59) were located at EZN (lat: 39°11′57.3″, long: 8°44′22.5″) in three collective parks. Nutritional management consisted of *ad libitum* hay and a commercial feed concentrate (1 kg per day per buck) to ensure an appropriate body condition score.

The flock was regularly screened for brucellosis and classified as brucellosis-free each year. Deworming and vaccination against clostridial and pasteurellosis diseases were performed twice a year.

These bucks were part of the BPAG ex situ conservation program, which was supported by specific projects and rural development programs over the years. Each animal was kept at EZN as a semen donor until reaching 250 doses (frozen straws) as a minimum requirement. Only ejaculates (n = 1017) with individual motility (IM) ≥ 30% and normal morphology (NM) ≥ 30% at thawed SPZ evaluation were approved for storage in our national gene bank (BPAG) and used in the present study.

### 2.2. Semen Collection, Evaluation, and Processing

Semen was collected by the artificial vagina technique coupled to a graduated glass tube. All collected ejaculates were placed in a water bath at 30 °C. Quantitative semen parameters (volume and concentration) were evaluated by direct observation (graduated test tube) and a calibrated spectrophotometer using a sperm dilution of 1/400, respectively. IM was evaluated in 10 µL of diluted semen (1/100 in saline solution; [12]) by estimating 5 microscopic fields (200× magnification) and estimating the SPZ% with progressive, fast, and straight movements. IM was subjectively evaluated by the same research group (three operational researchers) throughout the study period.

Good-quality ejaculates based on IM > 65% and semen concentration > 2 × 10^6^/mL were collected and processed. The IM evaluation in fresh semen was carried using an optic microscope (Olympus BX40 microscopic^®^, Tokyo, Japan) as above, a semen concentration of 10 µL diluted semen (10 µL of pure semen in 3990 µL of pure water (Milli Q^®^), and a spectrophotometer. After this preliminary evaluation, individual ejaculates were diluted with Krebs–Ringer phosphate glucose (KRP) solution to obtain a concentration of 800 SPZ × 10^6^/mL, and centrifuged at 800× *g* for 15 min to remove seminal plasma [13,14]. A second centrifugation was performed after adding a new volume of KRP, equal to the seminal plasma removal of each ejaculate. Afterwards, the semen pellet was finally diluted with a specific semen cryopreservation EZN extender to 800 SPZ × 10^6^/mL [15], and IM was checked again. In our experimental reproductive laboratory, our EZN extender contained 7% egg yolk and was used to store the semen at the gene bank (BPAG).

Smears with diluted semen (1/100) in saline physiologic solution and eosin–nigrosine (vital stain) were performed to ascertain sperm viability (live SPZ%), normality (NM%), and abnormalities (head defects, Head Ds; midpiece defects, MPDs; tail defects, Tail Ds) counting 100 SPZ under a 1000× magnification. Unstained SPZ presenting membrane integrity were classified as live SPZ, while those with pink/red color, meaning a damaged membrane, were classified as dead. Overall, damage to acrosome membrane integrity, the absence of an acrosome, head physical defects, and all membrane damage in the head of SPZ were classified as Head D; the presence of a double midpiece, proximal and distal cytoplasmatic droplets, and SPZ without a head were included in MPD; damage to the tail membrane and a ruptured, broken, bent, or thick tail were considered as Tail D.

Semen extended with the EZN extender was packed into 0.25 mL Cassou French straws^®^ (200 × 10^6^ SPZ/straw). The sealed straws were then placed inside a double boiler, at 28 °C, and placed in a refrigerated chamber (4 °C) for 4 h. During this period, a suitable gradual and balanced cooling of semen was performed. Afterwards, refrigerated straws were settled in a floating freezing rack (Minitube^®^) inside a stainless-steel box that contained 7 L of liquid nitrogen. These straws were placed 4 cm above the liquid nitrogen level, being frozen by nitrogen vapors (−120 °C) in a cryo-chamber (Minitube^®^) for 20 min. Then, the Cassou straws were submerged in liquid nitrogen and stored in containers.

Forty-eight hours after freezing, one straw from each ejaculate was thawed (37.5 °C for 50 s.) for SPZ re-evaluation. Afterwards, each straw was carefully cleaned and dried, and its contents were poured into a glass tube with 1 mL of saline solution (0.5%), which was placed in a water bath at 38 °C. Posteriorly, its content was slowly homogenized. Five minutes later, a sample (10 µL) of diluted semen was placed on a slide covered with a coverslip and placed over a heated microscope stage (37.5 °C). Diluted semen was evaluated as previously described for fresh semen.

### 2.3. Breeding and Non-Breeding Season

The breeding season was considered to be between September and January as defined for high (>45°) latitudes [16]. The “classical” non-breeding season was subdivided into two periods: from February to April, representing a short non-breeding season in goats (deep anestrus, such as described for latitudes about 39–42° N, and from May to August, where native goats can be naturally mated [6,15].

### 2.4. Statistical Analysis

A total of 1017 ejaculates, with complete and incomplete records (mostly from Preta de Montezinho breed), were obtained from all four goat breeds between 2002 and 2020. These data were used for the descriptive analysis of ejaculate and SPZ traits to maximize the sample size and characterize all breeds.

For SPZ trait analysis, complete records were obtained from 30 Serrana, 15 Bravia, 11 Charnequeira, and 3 Preta de Montezinho bucks that were used in a total of 601, 194, 167, and 23 sessions of semen collection, respectively.

The Kolmogorov–Smirnov test for normality was used for all the ejaculate and SPZ traits. These semen traits were not normally distributed, and they were best normalized by a square root transformation (VE, SPZ concentration, and total SPZ) and by an arcsine square root transformation (live SPZ, IM, SPZ, and morphological traits).

A mixed linear model with repeated measures, using the restricted maximum likelihood (REML) method, was built for each reproductive trait (live SPZ, IM, NM, Head D, MPD, and Tail D) following the equation:Y_ijmo_ = H_i_ + L_j_ + B_o_ + (HxL)_ij_ + (HxB)_io_ + (LxB)_oj_ + (HxLxB)_ijo_ + t_mi_ + e_ijmo_
where

Y_ijmo_ is a vector of all observations and represented by the least square value;

H_i_ is the fixed effect for breed (4 levels considering all the breeds);

L_j_ is the fixed effect for season (3 levels: September to January, February to March, and May to August);

B_o_ is the fixed effect for semen processing (2 levels: fresh and thawed semen);

(HxL)_ij_, (HxB)_io_, and (LxB)_jo_ are the two-way interactions;

(HxLxB)_ijo_ is the three-way interaction;

T_mi_ is the random effect for animal (m) within the breed;

e_ijmo_ is a vector of residuals.

For VE, SPZ concentration, and total number of SPZ, only fresh semen was used. The fixed effect for semen processing and the three-way interaction were removed from the general equation.

The Tukey test was used to test the differences of fixed factors and the Wald test to estimate the variance component of the mixed factor.

The correlation coefficients between the various semen traits were estimated using PROC CORR of software SAS^®^ 9.4, and the proportion of explained variability of each frozen semen trait was estimated by regression with PROC REG and forward selection with the same statistical software.

The statistical analysis of the data was performed using the statistical software SAS^®^ 9.4 (SAS Institute Inc., Cary, NC, USA, 2019). All the results, except descriptive data, were presented as least-squares means ± (Sqrt) SEM or 95% confidence interval (95% CI) for a 0.05 level of significance.

## 3. Results

### 3.1. Ejaculate Traits (Volume, Spermatozoa Concentration, and Total Sperm)

A total of 1017 ejaculates from 59 bucks were obtained throughout the studied period. The descriptive statistics of VE, SPZ concentration, and total SPZ per ejaculate according to each breed are reported in Table 1.

Overall, the volume and total SPZ per ejaculate were influenced by the buck breed and season. The SPZ concentration per ejaculate was influenced by the season remaining similar among breeds. Nonetheless, a breed × season interaction was observed (Table 2).

Overall, Bravia bucks produced less VE (0.55 ± 0.05 mL; *p* < 0.05) compared to Charnequeira (0.64 ± 0.06 mL), Preta de Montezinho (0.69 ± 0.11 mL), and Serrana (0.73 ± 0.03 mL) breeds. Considering all breeds, the VE was lower during February–April (deep anestrus) (0.54 ± 0.05 mL) and May–August (0.69 ± 0.04 mL) than in September–January (0.74 ± 0.04 mL; *p* < 0.05) (full breeding season). However, the variation pattern of this volume, mainly during the May–August period, was different among breeds according to the breed × season interaction. The SPZ concentration was lower in the full breeding season (*p* < 0.05) compared to the other two periods for all breeds, except for Preta de Montezinho bucks. Total SPZ per ejaculate remained similar (*p* > 0.05) between seasons in Bravia and Preta de Montezinho bucks, but it was higher (*p* < 0.05) toward full breeding season in Charnequeira and Serrana breeds. All values are reported in Table 3.

A significant effect (*p* < 0.01) of the animal on these three studied traits, which corresponds to repeatability, was observed in all the breeds. The REML variance component estimate of the animal variable represented 24.2%, 16.1%, and 13.2% of the total variance for VE, SPZ concentration, and total SPZ, respectively.

Overall (n = 889), significant positive correlations (*p* < 0.01) were observed between VE (r = 0.79) and SPZ concentration (r = 0.42) and total SPZ. Also, a negative significant correlation between VE and SPZ concentration (r = −0.18; *p* < 0.01) was found.

### 3.2. Spermatozoa Traits in Fresh and Thawed Semen

Overall, in fresh semen, the SPZ viability (live SPZ) was 75.0 ± 12.2%, IM 65.5 ± 5.7%, NM 85.8 ± 6.7%, HD 6.5 ± 4.8%, MPD 3.0 ± 3.4%, and Tail D 3.4 ± 3.8% (Table 4).

In frozen semen, the SPZ viability was 42.8 ± 10.7%, IM 42.9 ± 5.5%, NM 74.3 ± 9.6%, head defects 18.0 ± 9.4%, MPD 1.9 ± 2.2%, and Tail D 6.5 ± 4.0% (Table 5).

According to the mixed linear model, the freezing–thawing cycle was the main effect influencing (*p* < 0.01) all SPZ traits. The breed influenced (*p* < 0.05) the live SPZ and MPD. The season only affected (*p* < 0.05) the HD. However, some interactions between breed and season or semen processing were observed for some SPZ traits (Table 6).

Overall, the correlation of the SPZ traits between fresh and thawed semen was low (Table 7). However, the variability of SPZ viability, NM, and HD in thawed SPZ was mainly attributed to fresh SPZ. Inversely, IM variability was mainly explained by the VE (Table 8).

In fresh semen, a lower SPZ viability was observed in Serrana (74.7 ± 0.8%; *p* < 0.05) compared to Charnequeira (79.0 ± 1.4%) and Preta de Montezinho (80.5 ± 3.0%) bucks. However, in thawed semen, these differences only persisted between Serrana (43.3 ± 0.8%) and Preta de Montezinho (50.6 ± 3.3%; *p* < 0.05) breeds (Figure 1).

The IM was influenced (*p* < 0.01) by the freezing–thawing cycle and remained similar between breeds in thawed SPZ (Serrana = 45.5 ± 0.4%; Figure 2). However, in fresh semen, the IM was higher in Bravia (68.5 ± 0.6%) or Preta de Montezinho (70.6 ± 1.3%) than in Serrana (65.1 ± 0.4%; *p* < 0.05) bucks due to the breed × semen interaction (Figure 2).

Overall, the SPZ decreased with NM by 14.4% from 86.8 ± 1.0% (fresh semen) to 74.3 ± 1.0% (thawed semen; *p* < 0.01) after the freezing–thawing cycle. This decrease was mainly due to the HD increase (176.9%) from fresh (6.5%) to thawed (18.5%) semen. HDs were also influenced by season, and interactions between breed and semen or season. In thawed semen, more HDs were observed in Bravia (20.9 ± 1.0%) compared to Charnequeira (17.6 ± 1.1%; *p* < 0.05) (Figure 3). In fresh semen, the least HDs were observed in Bravia (5.5 ± 1.0%) and the most HDs in Serrana 6.8 ± 0.7%) bucks.

Overall, a lower percentage of HDs was observed in February–April (9.8 ± 1.4%; *p* < 0.05) than in May–August (13.7 ± 1.3%) or September–January (13.2 ± 1.1%), such as represented by Charnequeira bucks (Figure 4). However, this pattern differed for the other breeds. In Bravia and Serrana bucks, lower values were observed in both February–April (12.7 ± 1.1% and 10.6 ± 0.7%, respectively) and May–August (12.1 ± 1.3% and 11.9 ± 0.7%, respectively) than in September–January (14.9 ± 0.9% and 13.3 ± 0.7%, respectively). In Preta de Montezinho bucks, this pattern was inverse, in which the lowest values were observed in the full reproductive period (11.5 ± 2.7%).

The MPD decreased from an average of 3.0 ± 0.4% (fresh semen) to 1.9 ± 0.4% (thawed semen; *p* < 0.05) influenced by the freezing–thawing cycle (Figure 5). A similar pattern was observed in all breeds. Nonetheless, Bravia (3.1 ± 0.3%) had more MPDs compared to Preta de Montezinho (1.4 ± 0.8%; *p* < 0.05) bucks. This occurred mainly due to the most MPDs being observed during February–April (4.5 ± 0.3%; *p* < 0.05) in Bravia bucks (breed × season interaction).

On average, the Tail Ds were more numerous in thawed (6.5 ± 0.6%) than in fresh (3.4 ± 0.6%) semen (Figure 6). This pattern was observed in all breeds, except for Preta de Montezinho Bucks, which remained constant (3.4 ± 1.0%; breed × semen interaction; *p* < 0.05).

The REML variance component estimate of the animal variable represented 9%, 14%, 19%, 19%, 8%, and 7% for live SPZ, IM, NM, HD, MPD, and Tail D, respectively.

## 4. Discussion

### 4.1. Semen Production

Native Portuguese bucks produced on average 0.71 mL of ejaculate per session with an SPZ concentration of 4341 × 10^6^/mL. These values are within the range of the species [17,18,19]. In the present study, the lowest VE was observed in Bravia bucks, maybe related to the small size of the breed (35–50 kg in males) [3]. However, research on the comparative testicular volume in the Bravia breed is needed to confirm this aspect.

It is well known that semen production mainly varies according to testicular volume/weight, which varies depending on the season and testosterone levels [20,21,22].

Overall, the VE was highest during September–January for native Portuguese bucks. However, the degree of this influence seemed to be lower than that observed for other buck breeds located at high latitudes (e.g., Alpina and Saanen in France), which are still fully unproductive during the non-breeding season (February–August) [23,24,25]. In fact, a breed × season interaction in VE was observed in our study. A progressive increase in VE from February–April to September–January was observed in Serrana bucks, but May–August remains a transient-like period in Bravia and Charnequeira males. Nonetheless, the SPZ concentration was higher during September–January than in the other two periods for these three breeds. This aspect is consistent with the classical length of the non-breeding period previously reported for high-latitude goat breeds [19,26,27]. The sample size of Preta de Montezinho was smaller and results should be carefully interpreted and confirmed in further research. A reasonable amount of genetic and permanent environmental individual variability was observed for VE and other semen traits according to the REML variance component estimate of the animal variable in our mixed model (animal variable as random effect). However, the inclusion of the animal variable as the random effect allows for increasing the efficiency of the estimation of the whole model.

Compared to native Portuguese rams, this seasonal pattern of buck semen production is different, in which no seasonal variation in volume of ejaculate or SPZ concentration is observed [6]. It seems that sheep males are less sensitive to seasonality [28].

In the present study, the VE was strongly correlated (r = 0.79) with total SPZ per ejaculate. This finding indicates that especially Serrana and Charnequeira bucks are able (under natural conditions) to produce more total SPZ per ejaculate during the full breeding season (see Table 3), or even during May–August (Serrana breed). This aspect may have implications in flock reproductive management (e.g., male/female ratio) or male hormone use (e.g., melatonin) regarding the breed [19]. On the other hand, the VE was lowly (r = −0.18) correlated with SPZ concentration, allowing for the collection of semen throughout the year, in accordance with other studies on Mediterranean breeds [27].

### 4.2. Spermatozoa Traits in Fresh Semen

Overall, the values of SPZ traits remained similar between our native breeds (see Table 6). The breed effect on SPZ viability and MPD, or its interaction with season and semen on some SPZ traits, should be interpreted with caution. Most of these potential effects were due to the inclusion of the Preta de Montezinho breed, and its limitations, such as those reported before for VE. In fresh semen, these values are in agreement with other goat breeds worldwide with regard to the genotypic origin of the breed. In a study conducted in Murcia (Spain) on six Murciana-Granadina bucks (a native breed) counting 63 ejaculates, the total motility (TM; i.e., similar to the IM of our study) of SPZ in fresh semen was 64.2 ± 7.1% (±SE) with an IM of 3.92 ± 0.2 classified on a 0–5 scale [29]. A similar pattern of progressive motility (PM, i.e., SPZ progressing forward) using computer-assisted sperm analysis (CASA) was reported in seven Beni Arouss bucks (native breed from Morroco) with SPZ viability reaching more than 80% [30]. In Blanca de Rasquera bucks (Spanish native breed; Barcelona, Spain), the SPZ viability was 76.8 ± 4.3% [31]. In China, the PM of eight Xinong Saanen Bucks (originated as French breed) varied between 70.5 ± 3.2% in winter and 83.8 ± 1.6% in autumn (*p* < 0.05) [32].

#### 4.2.1. Effect of Season on Spermatozoa Traits

Contrarily to VE and SPZ concentration, the season did not exert relevant effects on SPZ traits. Only SPZ HDs were affected by season. The lowest proportion of HDs was observed in February–April for Charnequeira, and in May–August for Bravia and Serrana buck breeds. These different patterns of seasonal effects within breeds are interesting and should be addressed in further research. Nonetheless, native bucks can be year-round semen donors due to the high SPZ traits values. Interestingly, poor VE and kinematic SPZ traits were observed in Xinong Saanen Bucks using CASA during winter season at 34 N° latitude [32], but with individual variations. This period corresponds to the late full breeding and early non-breeding seasons.

#### 4.2.2. Effect of Freezing–Thawing Cycle on Spermatozoa Traits

Sperm cryopreservation and thawing procedures may strongly impair sperm function. Cryopreservation has a negative effect on acrosomal morphology, motility, and acrosome enzyme activity in sperm cells [33]. Overall, the net effect of cryopreservation is the loss of sperm-fertilizing capacity [34].

In the present study, a significant decrease (*p* < 0.05) in SPZ viability (from 75.0 to 42.8%) and IM (from 65.5 to 42.9%) was observed after the freezing–thawing cycle, which is in agreement with results previously obtained by our group for the Serrana breed [4]. This decrease varies according to other worldwide (native) breeds, but in general, it is similar to other European native breeds of the Mediterranean basin. Using the traditional method of subjective evaluation, Valencia et al. [35] reported a PM of 59.0% after thawing in Anglo-Nubia, alpine, and Saanen bucks (Mexico), and Vallecillo et al. [36] observed an IM of 60.5% in white Andalusian Serrana bucks (Spain), both using an egg-yolk-based semen extender. Both latter studies presented higher values of PM/IM compared to our study, maybe influenced by operators and other factors (e.g., goat breed, season, climate, etc.). Using the CASA system, Tabarez et al. [37] observed a TM and PM of 46.8% and 18.7%, respectively, using 15% powder egg-yolk-based semen extender from ejaculates of Blanca de Rasquera bucks. Other researchers [28] have observed a great variation in TM (18.3–56.7%) and PM (10.4–36.7%) explained by semen extender, goat breed, individual variation, and time-dependent motility decrease after thawing. Usually, these values decrease over time, being observed 13.7–62.3% (TM) and 6–41.6% (PM), 5 h after thawing. The IM evaluation after thawing is a good indicator of sperm quality, but it is essential that sperm cells remain viable, non-reactive, and progressively motile for 1 to 5 h after insemination to achieve acceptable fertility or to be selected for other artificial reproductive technologies.

SPZ degradation is largely independent of fresh semen material. In fact, according to our study, fresh semen is only responsible for 7.2% of SPZ viability and 4.4% of TM degradation in thawed SPZ. According to the partial R-squared values (see Table 8), the volume of ejaculate and SPZ concentration exerted a low effect on these traits. The sperm cryopreservation success is influenced by several factors, namely fresh semen quality, extender, freezing–thawing protocol, and evaluation method [4].

Semen from small ruminants is extremely sensitive to cryopreservation compared to other species [38]. Moreover, significant differences in semen freezability amongst breeds and males of the same species were clearly demonstrated [29,38]. Sperm cryopreservation induces damage in sperm cells, namely in the plasma membrane and acrosome integrity, causing a significant increase in dead SPZ and abnormalities. Acrosome integrity is a determinant factor of fertilization ability. The acrosome contains several enzymes that are essential for the oocyte–sperm interaction, namely sperm acrosome enzyme and hyaluronidase enzyme. Sperm motility was positively correlated with the activity of the two acrosome enzymes before and after fertilization [39]. One type of acrosomal damage is the leakage of enzymes that harm sperm fertility potential, and sometimes acrosome damage is related to precocious enzyme release [40].

A significant variation between fresh and frozen semen quality among donors has been identified. In fact, there are animals whose sperm freeze well (“Good Freezers”) while others present almost total necrospermia after thawing [29]. Several specific tests may be used to check sperm quality, namely motility, acrosome structure, enzyme activity, and fertilization ability. For instance, there were 5–6% damaged acrosomes and 85–90% normal SPZ in the fresh semen of Jamunapari Indian breed, compared to 38–43% and 48–53%, respectively, in frozen semen [41]. The fertilization ability of thawed semen from small ruminants is significantly decreased when compared to fresh semen due to a depression of SPZ motility and metabolism reflected in lower embryo production rates and quality [39].

For buck semen cryopreservation, it is essential to have a suitable extender capable of supporting several homeostatic conditions. According to Galián et al. [29], the use of a suitable semen extender is a determining factor for frozen semen quality, and can mean the success or failure of this biotechnology. Frozen semen of superior quality may be preserved in germplasm banks and used in breeding programs for rare and endangered species and breeds of domestic animals, as in the present study. In order to use egg yolk extenders, it is mandatory to remove the buck seminal plasma before semen dilution using a Krebs–Ringer phosphate solution associated with a double centrifugation [15]. Usually, the majority of semen extenders for buck sperm refrigeration or cryopreservation contain 15–20% egg yolk [31], whose lipoproteins have a protective effect on sperm membranes. Also, it contains 7% glycerol, as internal an cryoprotectant, which is mainly used for buck sperm cryopreservation [14,42].

In our study, a proportion of 6.5% HD was observed in fresh semen, which was similar to Jamunopari Indian bucks (>90% SPZ with normal head and acrosome morphology; [41]). However, a great SPZ increase in HDs was observed in our native bucks after the freezing–thawing cycle. An increase in HDs was observed in other ruminant species such as bucks [43], rams [44], and bulls [45]. A significant increase (*p* < 0.001) in amorphous (12.0 vs. 18.0%), megalo (7.5 vs. 10.5%), and elongated (6.5 vs. 9.0) heads of SPZ was also observed in humans after semen cryopreservation [46].

Capacitation is essential to sperm-fertilizing ability within the female reproductive tract and the cryopreservation process can initiate irreversible damage to sperm cells, leading to premature capacitation, which cause changes in sperm physiology and in fertilization itself [47]. Research has suggested that the lipid structure of bovine sperm is more resistant to cold stress than that of small ruminant sperm [39]. Sperm capacitation requires acrosome enzymes for sperm fertilization and are good predictors of sperm fertility potential. However, during cryopreservation, there are disturbances in plasma membrane and sperm membrane ultrastructure with the leakage of acrosomal enzymes [48], with significant decreases in sperm fertility in small ruminants, which is different from cattle. Also, when sperm are stored at low temperatures for long periods of time, enzyme activity, motility, and fertility are significantly reduced [49].

Moving spermatozoa tails suddenly become stuck and enclose the distal cytoplasmic droplet, becoming a sperm cell with a distal midpiece reflex (also called a hairpin-curved tail), or experience osmotic changes during freezing–thawing, which may be influenced by the extender composition, and can cause the bending of the tail at some of the distal-droplet-bearing sperm [11]. These authors observed that bent tails increased in the same percentage as distal droplets decreased in frozen semen compared to fresh semen. Also, normally, centrifugation decreases the number of distal droplets in bucks as in Collared Peccaries. In Jamunapari bucks, the ultrastructural changes were protrusion at the anterior cap, broken tail, swelling of the acrosome, and loss of acrosomal contents.

In the future, more sperm cryopreservation studies are needed to improve semen cryopreservation quality, fertility potential, and the reproductive outcomes of our native breeds.

### 4.3. Limitations of the Study

This study presents three main limitations. The first one is related to the subjective analysis of the SPZ traits. CASA is currently widely used as a reference to evaluate SPZ traits in domestic animals [50]. This methodology provides an automatic and objective measurements of viability, kinetic, and morphologic values of SPZ, among others. Nonetheless, in our study, the ejaculate collection and evaluation are performed by the same research group to preserve semen in the BPAG. This procedure ensured the normalization of all procedures.

The second one is related to the low sample size of some native breeds, namely the Preta de Montezinho as previously reported. Only three bucks were assessed, which cannot be representative of the respective population. Also, the number of selected ejaculates from this breed was low, affecting the statistical results.

The third limitation is related to the missing of some values in the records (see Table 4 and Table 5). This issue can cause slight differences in some results regarding the use of a complete database.

Nonetheless, these limitations were considered in the results interpretation. However, this is the largest study regarding the studied SPZ parameters in native Portuguese bucks, where straw semen samples were stored in the BPAG.

## 5. Conclusions

The VE, SPZ viability, IM, and morphology of four native Portuguese buck breeds were characterized in fresh and thawed semen. A lower VE and higher SPZ concentration were observed in Bravia compared to the other breeds. Also, more ejaculate and SPZ production was observed during the full breeding season (September–January), which was brought forward to May–Aug in some native breeds. However, in general, neither breed nor season had a relevant influence on the SPZ traits between breeds. The freezing–thawing cycle was the major significant cause of SPZ deterioration, but mainly related to extrinsic causes of fresh semen. Under a natural photoperiod, semen production and storage can be maximized during the breeding season, but the quality of SPZ traits remain stable for semen collection during the whole year.

## Figures and Tables

**Figure 1 vetsci-11-00326-f001:**
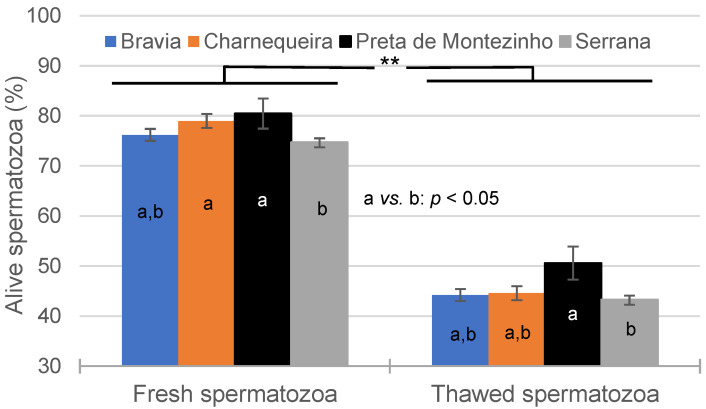
Effect of breed and semen cryopreservation on the spermatozoa viability (**: *p* < 0.01; a *vs.* b: *p* < 0.05).

**Figure 2 vetsci-11-00326-f002:**
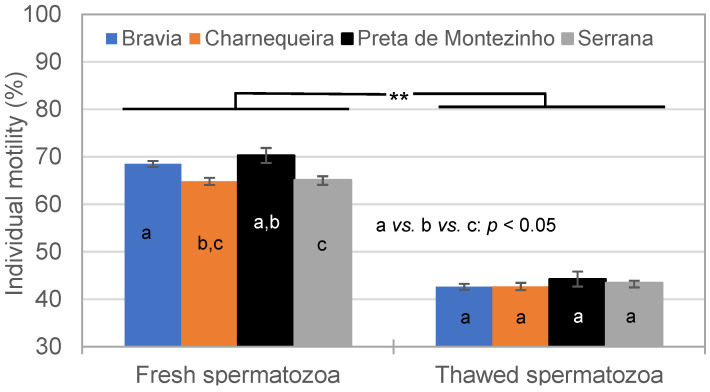
Effect of breed and semen cryopreservation on the individual motility (**: *p* < 0.01; a *vs.* b *vs.* c: *p* < 0.05).

**Figure 3 vetsci-11-00326-f003:**
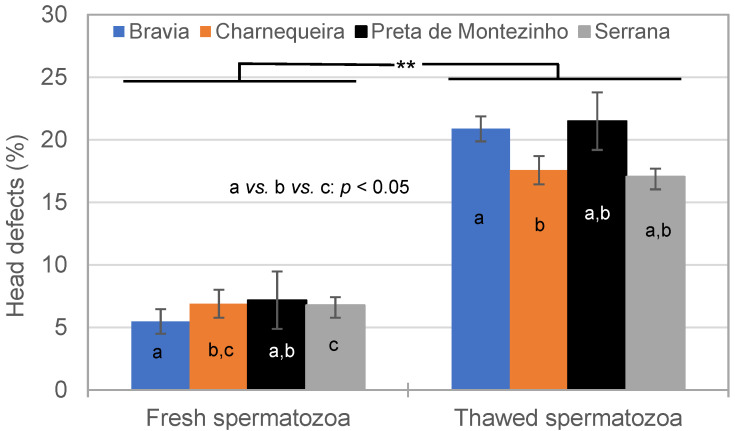
Effect of breed and semen cryopreservation on the head defects (**: *p* < 0.01; a *vs.* b *vs.* c: *p* < 0.05).

**Figure 4 vetsci-11-00326-f004:**
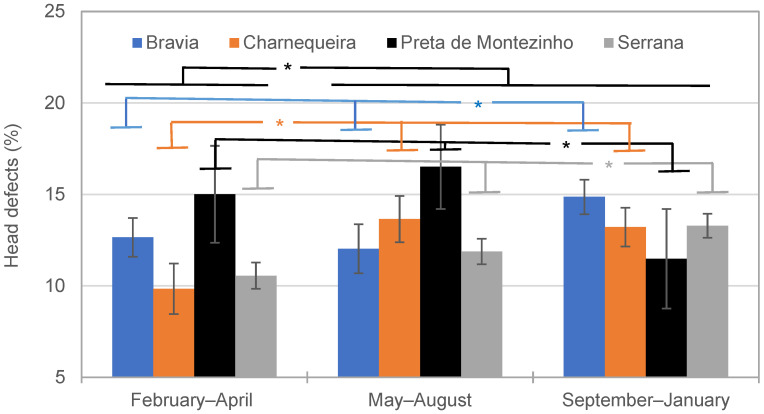
Effect of breed and season cryopreservation on the head defects (*: *p* < 0.05).

**Figure 5 vetsci-11-00326-f005:**
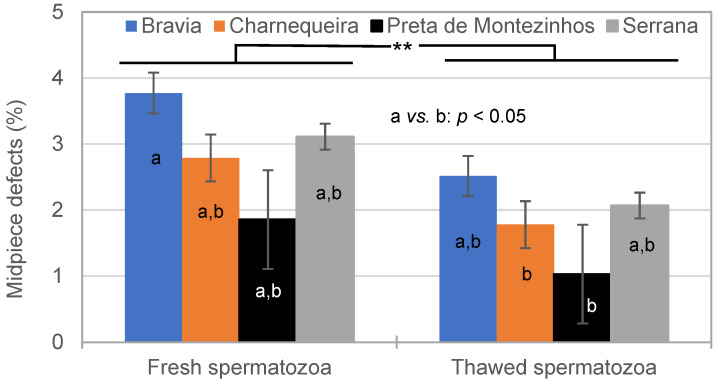
Effect of breed and semen cryopreservation on the midpiece defects (**: *p* < 0.01; a *vs.* b: *p* < 0.05).

**Figure 6 vetsci-11-00326-f006:**
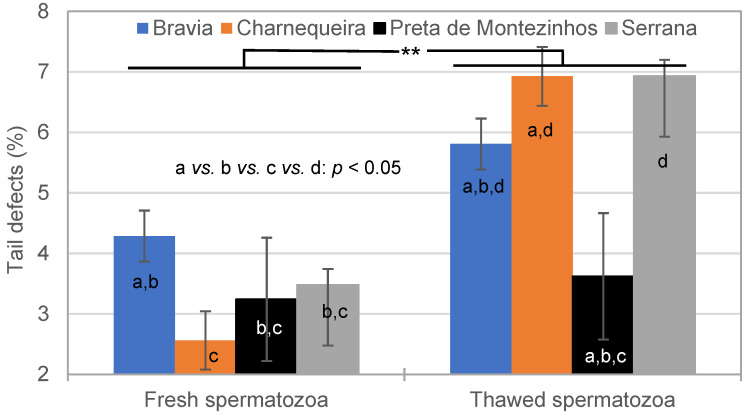
Effect of breed and semen cryopreservation on the tail defects (**: *p* < 0.01; a *vs.* b *vs.* c *vs.* d: *p* < 0.05).

**Table 1 vetsci-11-00326-t001:** Buck breeds, number of animals, and ejaculates obtained from 2002 to 2020 and respective descriptive analysis of ejaculate traits.

Breed	Number of Bucks	Number of Ejaculates	Ejaculate Traits								
			Volume (mL)			SPZ Concentration (10^6^/mL)			Total SPZ per Ejaculate (10^6^)		
			$\tilde{x}$	$\bar{x}$	CV%	$\tilde{x}$	$\bar{x}$	CV%	$\tilde{x}$	$\bar{x}$	CV%
Bravia	15	207	0.50	0.59	43	4340	4333	34	2375	2614	54
Charnequeira	11	171	0.68	0.72	47	4540	4585	29	3025	3311	52
Preta de Montezinho	3	23	0.70	0.70	48	5550	5307	17	5068	5181	20
Serrana	30	616	0.70	0.75	46	4150	4268	30	2790	3158	52
Total	59	1017	0.68	0.71	47	4300	4341	31	2750	3099	53

SPZ: spermatozoa.

**Table 2 vetsci-11-00326-t002:** Effects of breed and season, and breed × season interaction on volume, spermatozoa (SPZ) concentration, and total sperm per ejaculate on the four native breeds.

Traits/Effects	DF	Volume per Ejaculate (mL)	SPZ Concentration (10^6^/mL)	Total SPZ per Ejaculate (10^6^)
**Breed**	3	*	NS	**
**Season**	2	**	**	*
**Breed × Season**	3	*	*	NS
**N**	1017			

DF: degrees of freedom; *: *p* < 0.05, **: *p* < 0.01; NS: not significant.

**Table 3 vetsci-11-00326-t003:** Volume per ejaculate, spermatozoa (SPZ) concentration, and total sperm per ejaculate in the four native breeds at three different semen collection seasons (n = 1017).

Breed	Season	SPZ Traits (LSmean ± SEM)		
		Volume per Ejaculate (mL)	SPZ Concentration (10^6^/mL)	Total SPZ per Ejaculate (10^6^)
Bravia	February–April	0.47 ± 0.06 ^a^	5274 ± 245 ^a^	2480 ± 296 ^a^
	May–August	0.54 ± 0.07 ^a,b^	5052 ± 290 ^a^	2967 ± 352 ^a^
	September–January	0.65 ± 0.05 ^b^	3824 ± 180 ^b^	2537 ± 210 ^a^
Charnequeira	February–April	0.50 ± 0.08 ^a^	5226 ± 339 ^a^	2850 ± 412 ^a^
	May–August	0.61 ± 0.07 ^a^	4855 ± 249 ^a^	3121 ± 333 ^a^
	September–January	0.82 ± 0.06 ^b^	4327 ± 219 ^b^	3709 ± 254 ^b^
Preta de Montezinho	February–April	0.56 ± 0.15 ^a^	- *	- *
	May–August	0.87 ± 0.13 ^b^	5410 ± 556 ^a^	5049 ± 671 ^a^
	September–January	0.64 ± 0.15 ^a,b^	4922 ± 958 ^a^	5594 ± 1179 ^a^
Serrana	February–April	0.63 ± 0.04 ^a^	4723 ± 150 ^a^	3000 ± 178 ^a^
	May–August	0.74 ± 0.04 ^b^	4852 ± 141 ^a^	3541 ± 166 ^b^
	September–January	0.84 ± 0.04 ^c^	4017 ± 127 ^b^	3322 ± 147 ^b^

* The records of SPZ concentration and total SPZ per ejaculate were missed. ^a,b,c^ different superscript letters for each column and each breed: *p* < 0.05.

**Table 4 vetsci-11-00326-t004:** Descriptive statistics of spermatozoa (SPZ) traits (%) in fresh semen.

Breed	Bucks (n)	Ejaculates (n) *	Alive SPZ		IM		NM		HD		MPD		Tail D	
			$\bar{x}$	CV%	$\bar{x}$	CV%	$\bar{x}$	CV%	$\bar{x}$	CV%	$\bar{x}$	CV%	$\bar{x}$	CV%
Bravia	15	186–207	75.7	14.8	68.3	7.8	84.9	8.6	6.5	89.7	3.5	126.3	4.4	99.4
Charnequeira	11	164–170	77.6	14.4	64.5	7.9	88.4	6.9	6.0	68.2	2.6	115.9	2.7	110.0
Preta de Montezinho	3	22–23	80.8	8.9	70.3	5.9	86.8	4.8	7.5	49.8	2.0	118.8	3.3	60.1
Serrana	30	598–615	73.8	17.3	64.7	8.8	86.9	7.7	6.7	69.4	3.0	103.7	3.2	117.6
Total	59	970–1015	75.0	16.3	65.5	8.8	86.8	7.7	6.5	73.3	3.0	112.7	3.4	112.8

* Variation of samples regarding each SPZ trait (some records were missed). IM: individual motility; NM: normal morphology; HD: head defect; MPD: midpiece defect.

**Table 5 vetsci-11-00326-t005:** Descriptive statistics of spermatozoa (SPZ) traits (%) in thawed semen.

Breed	Bucks (n)	Ejaculates (n)	Alive SPZ (%)		IM (%)		NM (%)		HD (%)		MPD (%)		Tail D (%)	
			$\bar{x}$	CV%	$\bar{x}$	CV%	$\bar{x}$	CV%	$\bar{x}$	CV%	$\bar{x}$	CV%	$\bar{x}$	CV%
Bravia	15	174–206	43.6	24.0	42.4	12.9	71.4	16.1	22.2	48.5	2.3	107.3	5.9	64.8
Charnequeira	11	155–168	43.0	23.9	42.4	11.9	75.2	12.6	16.8	57.2	1.6	104.5	7.0	60.4
Preta de Montezinho	3	18–23	51.2	18.0	44.3	10.3	72.8	12.8	21.8	40.9	1.1	136.8	3.7	53.3
Serrana	30	553–612	42.3	25.7	43.1	13.2	74.9	11.7	17.0	50.0	2.0	113.2	6.7	60.7
Total	59	944–1009	42.8	25.0	42.9	12.9	74.3	12.9	18.0	52.0	1.9	112.1	6.5	62.0

**Table 6 vetsci-11-00326-t006:** Effects of breed, season, and semen cryopreservation on spermatozoa (SPZ) traits (%).

Traits/Effects	DF	Alive SPZ	IM	NM	HD	MPD	Tail D
**Breed**	3	*	NS	NS	NS	*	NS
**Season**	2	NS	NS	NS	*	NS	NS
**Semen**	1	**	**	**	**	**	**
**Breed × Season**	6	NS	NS	NS	*	**	NS
**Breed × Semen**	3	*	**	NS	**	NS	**
**Season × Semen**	2	NS	NS	NS	NS	NS	NS
**Breed × Season × Semen**	6	NS	NS	NS	NS	NS	NS

IM: individual motility; NM: normal morphology; HD: head defect; MPD: midpiece defect; DF: degrees of freedom; *: *p* < 0.05, **: *p* < 0.01; NS: not significant.

**Table 7 vetsci-11-00326-t007:** Correlation coefficients between the same trait in fresh and frozen semen.

Traits	n	r	R-Squared (%)
Alive (%)	925	0.269 **	7.2
Individual motility (%)	1009	0.210 **	4.4
Normal morphology (%)	938	0.274 **	7.5
Head defects (%)	931	0.290 **	8.4
Intermediate piece defects (%)	983	0.330 **	10.9
Tail defects (%)	892	0.066 ^NS^	0.4

Significance level: ** *p* < 0.01; NS: not significant.

**Table 8 vetsci-11-00326-t008:** Proportion of explained variability of each thawed spermatozoa (SPZ) trait estimated by regression and forward selection.

	Total R-Squared	Partial R-Squared (*p* < 0.05)			
Traits		Trait in Fresh SPZ	Volume per Ejaculate	SPZ Concentration	Total SPZ
Live (%)	0.099	0.060	0.030	0.009	-
Individual motility (%)	0.091	0.030	0.061	-	-
Normal morphology (%)	0.089	0.081	-	0.008	-
Head defects (%)	0.112	0.106	-	0.006	-
Midpiece defects (%)	0.136	0.115	0.012	0.008	-
Tail defects (%)	-	-	-	-	-

## Data Availability

The data present in this study are contained within this article.

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
