# Peer review of "Effect of Breed and Season in Buck Semen Cryopreservation: The Portuguese Animal Germplasm Bank"

_vetsci, 2024, doi:10.3390/vetsci11070326_

Round 1
Reviewer 1 Report
Comments and Suggestions for Authors
The manuscript analyzes the effect of breed, season and semen processing and their interaction on buck sperm parameters. The data has been acquired over a period of sixteen years, which makes the study interesting from a time perspective. During this long-term study the authors have collected 1017 ejaculates from different individuals belonging to the Serrana, Bravia, Charnequeira and Preta de Montezinho native portuguese goat breeds. Also, data are divided in three reproductive periods (Febr-Apr; May-Aug; Sep-Jan). The sperm parameters assessed are the ejaculate volume and concentration and the sperm viability, motility and morphology.
Questions about the study:
-Line 150: please specify here the cryopreservation extender that was used (it is mentioned later in line 494).
-Line 158-159: do the authors mean the straws were put inside a double boiler?
-It is not mentioned if the interaction sperm processing x breed is non-significant.
-Lines 166-171: there is a meaningful difference between thawing one or two straws. Thawing a couple of straws minimizes variability on results and accounts for possible differences derived from the procedure of thawing and freezing. Also, it is not clear from the text if, when thawing two straws, both straws were poured in the same tube or in a different one.
-Lines 119, 183, 228: different periods of ejaculate collection are stated, which is confusing (2003 to 2019; 2004 to 2020; 2002 to 2019?).
-Line 260: the random effect for animal within a breed affects the ejaculate volume, sperm concentration and total sperm per ejaculate, but not the sperm viability. Because the authors discuss about the existence of good freezer sperm donors (lines 479-481) it could be expected that a percentage of the variance in viable sperm cells was affected by differences among donors. This effect could be particularly obvious in the Preta de Montezinho breed with only three individuals. The authors are encouraged to discuss about this fact further, and also about the convenience of establishing individuals as random factors instead of fixed ones in studies with a small sample.
-Figures 1 and 2. The Serrano breed shows worse quality for some sperm parameters than the Preta de Montezinho breed. Could these results have been affected by the difference in sample size between both breeds? The authors are already aware that their data should be carefully interpreted (line 395 and 412).
Comments on the Quality of English Language
The manuscript is well-written but some minor spelling errors have been found, among which the following:
-Line 18: “Aug” instead of “Ago”
-Line 22: “(Preta de Montezinho) buck ejaculates”
-Line 55: move punctuation sign after “(meat purpose)”
-Line 59: “purpose” instead of “propose”
-Line 155: “damaged”
Please, revise for other misspellings.
Author Response
-Line 150: please specify here the cryopreservation extender that was used (it is mentioned later in line 494).
Moved (L153-155).
-Line 158-159: do the authors mean the straws were put inside a double boiler?
The sentence was improved to: “… placed inside a double boiler, at 28 ºC…”. (L167).
-It is not mentioned if the interaction sperm processing x breed is non-significant.
The interaction sperm processing x breed (“Breed x Semen”) is specifically reported in Table 6 for all SPZ parameters. If the review judge more adequate, we can change the name of the variable “semen” to “semen processing”.
-Lines 166-171: there is a meaningful difference between thawing one or two straws. Thawing a couple of straws minimizes variability on results and accounts for possible differences derived from the procedure of thawing and freezing. Also, it is not clear from the text if, when thawing two straws, both straws were poured in the same tube or in a different one.
The sentence was clarified: “… one straw…”. (L174).
Only one straw was randomly selected used for each evaluation. In some few cases, due to troubles during thawing process or other, the straw was rejected and a second straw was thawed.
-Lines 119, 183, 228: different periods of ejaculate collection are stated, which is confusing (2003 to 2019; 2004 to 2020; 2002 to 2019?).
Thanks for this advice. The period of the study was defined consistently (2002 to 2020). This confusion is due that not all breeds are used in all years. The change was made when due (L119, 191, 236).
-Line 260: the random effect for animal within a breed affects the ejaculate volume, sperm concentration and total sperm per ejaculate, but not the sperm viability. Because the authors discuss about the existence of good freezer sperm donors (lines 479-481) it could be expected that a percentage of the variance in viable sperm cells was affected by differences among donors. This effect could be particularly obvious in the Preta de Montezinho breed with only three individuals. The authors are encouraged to discuss about this fact further, and also about the convenience of establishing individuals as random factors instead of fixed ones in studies with a small sample.
This is a relevant question. The low sample size in Preta de Montezinho breed had a important effect (we tested it), and using animal variable as random factor can improved the estimation of some outcomes:
L407-412: “A reasonable amount of genetic and permanent environmental individual variability was observed for VE and other semen traits according to of REML variance component estimate of animal variable. our mixed model (animal variable as random effect). However, the inclusion of animal variable as random effect allows to increase the efficiency of estimation of the whole model.”.
-Figures 1 and 2. The Serrano breed shows worse quality for some sperm parameters than the Preta de Montezinho breed. Could these results have been affected by the difference in sample size between both breeds? The authors are already aware that their data should be carefully interpreted (line 395 and 412).
Partially, yes. But we don’t kwon the degree of this influence. Please note in Figure 1 and 2 (and others) that significance of differences between Preta de Montezinho breed and other breeds is variable (significant and not significant). So, other factors (not identified in this study) also can affect the results. Also, the variance is always highest in Preta de Montezinho bucks, usually related to small samples. If possible, we prefer to take some precaution in the interpretation of results regarding the effect of the sample size. This is very small population, and more bucks need be evaluated.
The manuscript is well-written but some minor spelling errors have been found, among which the following:
-Line 18: “Aug” instead of “Ago”
Corrected, thanks.
-Line 22: “(Preta de Montezinho) buck ejaculates”
Corrected, thanks.
-Line 55: move punctuation sign after “(meat purpose)”
Corrected, thanks.
-Line 59: “purpose” instead of “propose”
Corrected, thanks.
-Line 155: “damaged”
Corrected, thanks.
Please, revise for other misspellings.
Done

Reviewer 2 Report
Comments and Suggestions for Authors
Specific comments, critiques and suggestions for improvement are below.
Line 55: Remove the period (.) before the word except.
Lines 95 to 99: This sentence is confusing to this reviewer, should it say “The two major procedures that should be implemented are the characterization of ejaculates and spermatozoa (SPZ) traits in 1) fresh semen to be used for natural matting regarding the different seasons of semen collection of native goat breeds, and 2) after freezing-thawing cycle for artificial insemination purposes.”?
Line 97: Should ‘matting’ be ‘mating’?
Line 122: Should it be “(1 kg per day per buck)” instead of “(1 kg per day and buck)”?
Lines 138 to 140: Was individual motility (IM) evaluated using a Computer-Assisted Semen Analysis (CASA) system? Or was IM evaluated using traditional subjective human evaluation (i.e. no CASA system)? If human evaluation was performed, how many human evaluators performed the analysis? Is this question addressed at the end of the manuscript in ‘4.3. Limitations of the study’?
Line 142 to 143: Please provide a description of the microscope used.
Line 153: Please provide a description of the abnormalities (head defects, head D; midpiece defects, MPD; Tail defects, Tail D). For example, what constitutes a midpiece defect?
Line 228: Should 2019 be 2020? In line 119, it says that semen collection occurred between 2004 and 2020.
Line 261: REML? (not RELM?)
Line 265: What does n=889 refer to?
Line 337: Was the highest HD in Serrana bucks? To this reviewer, it appears that the black column (Preta de Montezinho) is higher than the gray column (Serrana).
Figures 5 and 6: Please be consistent in presentation of these figures compared with the previous figures. In figures 5 and 6, the Thawed spermatozoa are shown on the left side and the Fresh spermatozoa are shown on the right side. This is opposite for what is shown in figures 1, 2 and 3, for example.
Line 373: REML? (not RELM?)
Line 366: ‘On’ average, not ‘In’ average.
Line 380: Can a citation be given to verify the statement “These values are within the range of the species.”
Line 388: Possibly re-word to “ . . . which are still fully unproductive . . .”
Line 393: Should the word ‘consistent’ be used instead of “consonant”?
Lines 404 to 405: This reviewer is not familiar with the term (di)suse. Is the parentheses used in the proper location here? Is it supposed to be (dis)use, which would indicate an improper use of melatonin?
Line 405: Perhaps use the word “lowly” instead of “little”.
Line 416: What does ‘TM’ stand for? This review apologizes for missing the definition of ‘TM’.
Line 418: What does ‘PM’ stand for? This review apologizes for missing the definition of ‘PM’.
Line 426: Exert, not exerted.
Line 462: Please re-word this sentence. Perhaps the sentence could be re-written as “The SPZ degradation is largely independent of fresh semen material.”
Line 481: Remove ‘and’after ‘Several’.
Line 482: Remove ‘upon’.
Line 515: Should the word ‘kwon’ be ‘known’?
Line 517: Use commas in this sentence, “ . . . morphology, namely acrosome area, has . . .
Line 520: Remove the space between ‘50’ and ‘%’.
Lines 543: Perhaps use the word ‘begin’ instead of ‘originate’.
Line 553: Use the words 'distal cytoplasmic’ in front of ‘droplet’.
Section 4.2.2. Effect of freezing-thawing cycle on spermatozoa traits
Overall, this section contains good information, but this part of the discussion section could be condensed to address the findings of the research more specifically. For example, this reviewer’s opinion is that the paragraph covered by lines 510 to 514 could be removed completely since it does not address characteristics that were evaluated in the present study. This sentiment also applies to lines 515 to 541 and lines 561 to 579.
Comments on the Quality of English LanguageMinor editing of English language required
Author Response
We would thank the editor and reviewers for their time and accurate review of this manuscript, which improved it final quality. All changes (requested or not) were highlighted (red color) in the main document, except for the changes of the Figures 5 and 6 (just the requested graphical adaptation), spaces and update on citations and references, to keep the revised version clean as possible.
Line 55: Remove the period (.) before the word except.
Corrected, thanks.
Lines 95 to 99: This sentence is confusing to this reviewer, should it say “The two major procedures that should be implemented are the characterization of ejaculates and spermatozoa (SPZ) traits in 1) fresh semen to be used for natural matting regarding the different seasons of semen collection of native goat breeds, and 2) after freezing-thawing cycle for artificial insemination purposes.”?
It was corrected. Many thanks.
Line 97: Should ‘matting’ be ‘mating’?
Corrected, thanks.
Line 122: Should it be “(1 kg per day per buck)” instead of “(1 kg per day and buck)”?
Corrected, thanks.
Lines 138 to 140: Was individual motility (IM) evaluated using a Computer-Assisted Semen Analysis (CASA) system? Or was IM evaluated using traditional subjective human evaluation (i.e. no CASA system)? If human evaluation was performed, how many human evaluators performed the analysis? Is this question addressed at the end of the manuscript in ‘4.3. Limitations of the study’?
IM was subjectively evaluated by the same research group (three operational re-searchers) throughout the study period. (L140-141).
L552-555: “CASA is a currently widely used as reference to evaluate SPZ traits in domestic animals [50]. This methodology provides an automatic and objective measurements of viability, kinetic and morphologic values of SPZ, among others.”
Line 142 to 143: Please provide a description of the microscope used.
(Olympus BX40 microscopic®, Tokyo, Japan). (L144).
Line 153: Please provide a description of the abnormalities (head defects, head D; midpiece defects, MPD; Tail defects, Tail D). For example, what constitutes a midpiece defect?
“Overall, damage in acrosome membrane integrity, absence of acrosome, head physical defects, and all membrane damages in head of SPZ were classified as Head D; the presence of double midpiece piece, proximal and distal cytoplasmatic droplets and SPZ without head were included in MPD; and, the damages in tail membrane, and ruptured, broken, bent or thick tail were considered as tail D.”. (L160-164).
And
Line 228: Should 2019 be 2020? In line 119, it says that semen collection occurred between 2004 and 2020.
The study period was corrected to 2002 and 2020; some breeds like Preta de Montezinho, only were used for a short time. The change was made when due (L119, 191, 236).
Line 261: REML? (not RELM?)
Corrected, thanks.
Line 265: What does n=889 refer to?
889 refers to the total number of records taken into account when calculating this correlation coefficient. 889 Ejaculates, SPZ concentration and total SPZ records were non-null.
Line 337: Was the highest HD in Serrana bucks? To this reviewer, it appears that the black column (Preta de Montezinho) is higher than the gray column (Serrana).
Regarding the absolute values, you are right. But, please not that the value of Serrana goats is statistically higher (P <0.05) than all the other breeds, in fresh semen.
Figures 5 and 6: Please be consistent in presentation of these figures compared with the previous figures. In figures 5 and 6, the Thawed spermatozoa are shown on the left side and the Fresh spermatozoa are shown on the right side. This is opposite for what is shown in figures 1, 2 and 3, for example.
Corrected, thanks.
Line 373: REML? (not RELM?)
Corrected, thanks.
Line 366: ‘On’ average, not ‘In’ average.
Corrected, thanks.
Line 380: Can a citation be given to verify the statement “These values are within the range of the species.”
The citations [17-19] were inserted. (L391).
Line 388: Possibly re-word to “ . . . which are still fully unproductive . . .”
Corrected, thanks.
Line 393: Should the word ‘consistent’ be used instead of “consonant”?
Corrected, thanks.
Lines 404 to 405: This reviewer is not familiar with the term (di)suse. Is the parentheses used in the proper location here? Is it supposed to be (dis)use, which would indicate an improper use of melatonin?
Now, we used only the term “use”. The seasonality and as consequence the use of melatonin can be distinct between breeds. (L420).
Line 405: Perhaps use the word “lowly” instead of “little”.
Corrected, thanks.
Line 416: What does ‘TM’ stand for? This review apologizes for missing the definition of ‘TM’.
“… total motility (TM; i.e., similar to the IM of our study) of SPZ in fresh semen was 64.2 ± 7.1% (±SE) with an IM of 3.92±0.2 classified in a 0-5 scale [29].”. There are different definitions between studies for kinetics values. (L432-435).
Line 418: What does ‘PM’ stand for? This review apologizes for missing the definition of ‘PM’.
“Similar pattern of progressive motility (PM, i.e., SPZ progressing forward) …”. (L434).
Line 426: Exert, not exerted.
Corrected, thanks.
Line 462: Please re-word this sentence. Perhaps the sentence could be re-written as “The SPZ degradation is largely independent of fresh semen material.”
Corrected, thanks.
Line 481: Remove ‘and’after ‘Several’.
Corrected, thanks.
Line 482: Remove ‘upon’.
Corrected, thanks.
Line 515: Should the word ‘kwon’ be ‘known’?
Not applicable.
Line 517: Use commas in this sentence, “ . . . morphology, namely acrosome area, has . . .
Not applicable.
Line 520: Remove the space between ‘50’ and ‘%’.
Not applicable.
Lines 543: Perhaps use the word ‘begin’ instead of ‘originate’.
Corrected, thanks.
Line 553: Use the words 'distal cytoplasmic’ in front of ‘droplet’.
Corrected, thanks.
Section 4.2.2. Effect of freezing-thawing cycle on spermatozoa traits
Overall, this section contains good information, but this part of the discussion section could be condensed to address the findings of the research more specifically. For example, this reviewer’s opinion is that the paragraph covered by lines 510 to 514 could be removed completely since it does not address characteristics that were evaluated in the present study. This sentiment also applies to lines 515 to 541 and lines 561 to 579.
The paragraphs covered by lines 510 to 514, lines 515 to 541 and lines 561 to 579 were removed.
Comments on the Quality of English Language
Minor editing of English language required
Edition was done

Reviewer 3 Report
Comments and Suggestions for Authors
The manuscript “Effect of breed and season in buck Semen Cryopreservation: The Portuguese animal germplasm bank” aimed to characterize the semen as well as the influence of breed, season and semen processing on spermatozoa (SPZ) traits of four Portuguese native goats breeds used for the bank of Portuguese animal germplasm (BPAG).
In my opinion this is a classic example of a good paperwork. The whole manuscript is complete and intelligible. The authors address a simple question with excellent statistical methods. This gives the results meaning beyond statements about buck goat sperm. Text, table and figures are well balanced. The relevant literature has been considered. The authors have carefully indicated the limitations of the study.
Below, you can find my specific comments to the individual parts of the manuscript:
L121: indicate in italics “ad libitum”
L138: Sperm preparation medium (saline solution) is not frequently used for goat sperm? Why authors used this medium. Is there any information available on comparisons with other more commonly used media? Unpublished lab data or studies? please confirm it.
L141: In this study authors proposed a sperm motility analysis by direct observation as an alternative to existing conventional sperm analysis methods. However, CASA is still considered as robust and the most reliable system for analysis of sperm motility.
Author Response
L121: indicate in italics “ad libitum”
Corrected, thanks.
L138: Sperm preparation medium (saline solution) is not frequently used for goat sperm? Why authors used this medium. Is there any information available on comparisons with other more commonly used media? Unpublished lab data or studies? please confirm it.
Thanks to apport this interesting issue. The use of saline solution to evaluate IM and other kinetic sperm parameters are widely used in our laboratory (please, see https://doi.org/10.3390/ani11092619) and worldwide for ruminants species (e.g., https://doi.org/10.5713/ab.22.0184; https://doi.org/10.46419/vs.54.2.1, http://dx.doi.org/10.14202/vetworld.2020.840-846 )
We have added a specific citation [12] for bucks. (L138).
L141: In this study authors proposed a sperm motility analysis by direct observation as an alternative to existing conventional sperm analysis methods. However, CASA is still considered as robust and the most reliable system for analysis of sperm motility.
The reviewer is full right.
“CASA is a currently widely used as reference to evaluate SPZ traits in domestic animals [50]. This methodology provides an automatic and objective measurements of viability, kinetic and morphologic values of SPZ, among others.”. (L552-555).

Round 2
Reviewer 1 Report
Comments and Suggestions for Authors
The authors have addressed all the questions proposed by the reviewer and there are no further comments.
Reviewer 2 Report
Comments and Suggestions for Authors
Thank you for the kind responses to the suggestions and comments.
Comments on the Quality of English LanguageMinor editing of English language required
Reviewer 3 Report
Comments and Suggestions for Authors
The authors replied to most of the reviewer's questions.
The enormous effort made by the authors to improve the manuscript is noteworthy.